# Comprehensive phylogenetic analysis of the ribonucleotide reductase family reveals an ancestral clade

**Andrew A Burnim[1†], Matthew A Spence[2†], Da Xu[1†], Colin J Jackson[2,3,4]\*, Nozomi Ando[1]\***

[1]Department of Chemistry and Chemical Biology, Cornell University, Ithaca, United States; [2]Research School of Chemistry, Australian National University, Canberra, Australia; [3]Australian Research Council Centre of Excellence for Innovations in Peptide and Protein Science, Australian National University, Canberra, Australia; [4]Australian Research Council Centre of Excellence in Synthetic Biology, Australian National University, Canberra, Australia

**Abstract** Ribonucleotide reductases (RNRs) are used by all free-living organisms and many viruses to catalyze an essential step in the de novo biosynthesis of DNA precursors. RNRs are remarkably diverse by primary sequence and cofactor requirement, while sharing a conserved fold and radical-based mechanism for nucleotide reduction. Here, we structurally aligned the diverse RNR family by the conserved catalytic barrel to reconstruct the first large-scale phylogeny consisting of 6779 sequences that unites all extant classes of the RNR family and performed evo-velocity analysis to independently validate our evolutionary model. With a robust phylogeny in-hand, we uncovered a novel, phylogenetically distinct clade that is placed as ancestral to the classes I and II RNRs, which we have termed clade Ø. We employed small-angle X-ray scattering (SAXS), cryogenic-electron microscopy (cryo-EM), and AlphaFold2 to investigate a member of this clade from *Synechococcus* phage S-CBP4 and report the most minimal RNR architecture to-date. Based on our analyses, we propose an evolutionary model of diversification in the RNR family and delineate how our phylogeny can be used as a roadmap for targeted future study.

**\*For correspondence:**
colin.jackson@anu.edu.au (CJJ);
nozomi.ando@cornell.edu (NA)

[†]These authors contributed equally to this work

**Competing interest:** The authors declare that no competing interests exist.

## Editor's evaluation

The largest phylogeny of ribonucleotide reductases (RNRs) to-date is presented in this study via combining phylogenetic reconstruction, structural determination and analysis, and extensive discussion on the possible evolutionary history. A new clade of RNRs is uncovered and systematically compared with other well-known lineages of RNRs. Summarizing all observations and integrating previous knowledge, the authors propose a reasonable and not over-interpreted evolutionary model to depict the history of this ancient protein family. In sum, the arguments and data offered throughout this manuscript are strong and convincing, and they can help the community to comprehend the evolution of the diverse and crucial RNRs from a deep perspective.

## Introduction

The wealth of genomic and metagenomic sequence data that has exploded in recent years provides a new opportunity to reexamine enzyme families using large-scale and robust bioinformatic analyses. A particularly important enzyme family that necessitates such analyses are the ribonucleotide reductases (RNRs). RNRs are used by all free-living organisms and many viruses for the conversion of

**eLife digest** Billions of years ago, the Earth's atmosphere had very little oxygen. It was only after some bacteria and early plants evolved to harness energy from sunlight that oxygen began to fill the Earth's environment. Oxygen is highly reactive and can interfere with enzymes and other molecules that are essential to life. Organisms living at this point in history therefore had to adapt to survive in this new oxygen-rich world.

An ancient family of enzymes known as ribonucleotide reductases are used by all free-living organisms and many viruses to repair and replicate their DNA. Because of their essential role in managing DNA, these enzymes have been around on Earth for billions of years. Understanding how they evolved could therefore shed light on how nature adapted to increasing oxygen levels and other environmental changes at the molecular level.

One approach to study how proteins evolved is to use computational analysis to construct a phylogenetic tree. This reveals how existing members of a family are related to one another based on the chain of molecules (known as amino acids) that make up each protein. Despite having similar structures and all having the same function, ribonucleotide reductases have remarkably diverse sequences of amino acids. This makes it computationally very demanding to build a phylogenetic tree.

To overcome this, Burnim, Spence, Xu et al. created a phylogenetic tree using structural information from a part of the enzyme that is relatively similar in many modern-day ribonucleotide reductases. The final result took seven continuous months on a supercomputer to generate, and includes over 6,000 members of the enzyme family.

The phylogenetic tree revealed a new distinct group of ribonucleotide reductases that may explain how one adaptation to increasing levels of oxygen emerged in some family members, while another adaptation emerged in others. The approach used in this work also opens up a new way to study how other highly diverse enzymes and other protein families evolved, potentially revealing new insights about our planet's past.

ribonucleotides to 2'-deoxyribonucleotides in the *de novo* biosynthesis of DNA precursors (*Torrents et al., 2002*). RNRs are especially fascinating from an evolutionary perspective as they exhibit high diversity in primary sequence and cofactor requirement (*Lundin et al., 2015*), yet they share a common fold and radical-based catalytic mechanism for nucleotide reduction (*Licht et al., 1996*).

Based on current biochemical evidence, the full catalytic cycle is thought to involve three steps in all RNRs (*Figure 1—figure supplement 1*): cofactor-mediated generation of a thiyl radical at a conserved cysteine in the active site, nucleotide reduction, and re-reduction of the active site after turnover (*Greene et al., 2020*; *Holmgren and Sengupta, 2010*). Despite being a diverse enzyme family, the core structure of the catalytic subunit (known as α) is a conserved 10-stranded α/β barrel with the thiyl radical on the so-called 'finger loop' which connects the two halves of the barrel (*Figure 1*; *Uhlin and Eklund, 1996*). RNRs have been biochemically classified into three major groups based on the cofactor used to generate the thiyl radical (*Figure 1—figure supplement 1*). Class I RNRs use a ferritin subunit (β) to house a stable radical cofactor and are further subclassified by the metal content of the cofactor (*Cotruvo and Stubbe, 2011*; *Ruskoski and Boal, 2021*). For every turnover, the α and β subunits must form a complex to engage in long-range radical transfer to the active site (*Kang et al., 2020*; *Seyedsayamdost et al., 2007*). In class II RNRs, the thiyl radical is generated by a 5'-deoxyadenosyl radical (5'-dAdo•) produced from an adenosylcobalamin (AdoCbl) cofactor bound within the active site, and thus these enzymes do not require additional subunits (*Licht et al., 1996*). In class III RNRs, the thiyl radical is generated by a glycyl radical on the C-terminal domain of the α subunit, which itself is generated by a separate activase enzyme that utilizes *S*-adenosylmethionine (AdoMet) bound to a [4Fe-4S] cluster for radical chemistry (*Wei et al., 2014b*). Once the thiyl radical is produced in the active site, nucleotide reduction proceeds via a conserved mechanism for all classes (*Licht et al., 1996*; *Figure 1—figure supplement 1*). In most RNRs, nucleotide reduction is coupled with the oxidation of a conserved pair of active-site cysteines (*Figure 1—figure supplement 1*; *Booker et al., 1994*; *Mao et al., 1992*; *Wei et al., 2014a*), which are ultimately reduced by a thioredoxin system to enable the next round of turnover, although some class III RNRs use formate to directly reduce the active site (*Wei et al., 2014b*).

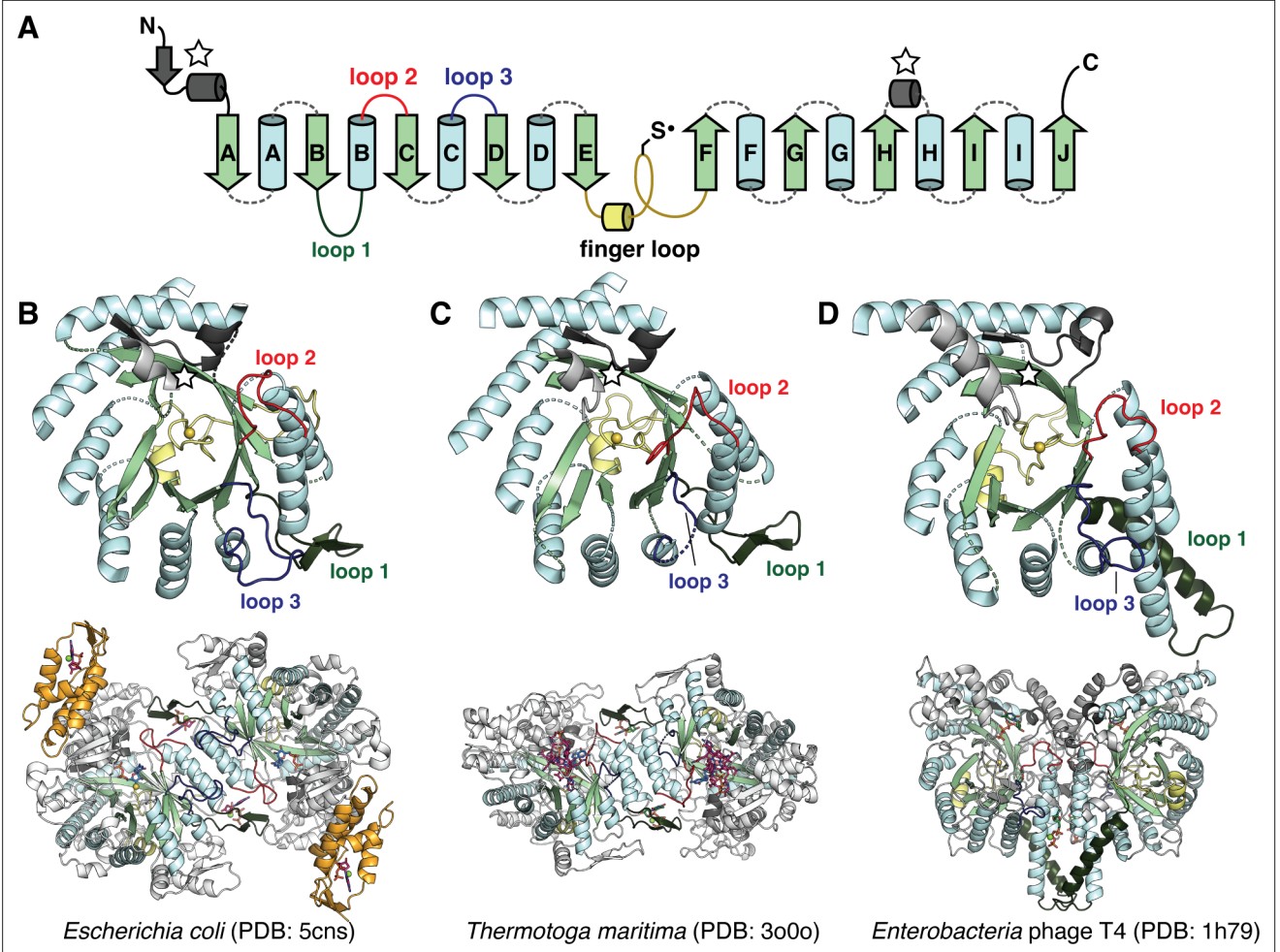

**Figure 1.** The catalytic fold of the ribonucleotide reductase (RNR) family is a unique 10-stranded α/β barrel, consisting of 10 β-strands (light green) and 8 α-helices (light blue). (**A**) Each half of the barrel contains a five-stranded parallel β-sheet (βA-βE and βF-βJ) that is arranged in anti-parallel orientation with respect to each other. The two halves are connected by the so-called 'finger loop' (yellow) which typically begins with a short α-helix and contains a conserved cysteine that has been shown to be the site of the catalytically essential thiyl radical in all biochemically characterized RNRs. Diversity among the RNRs is generated by N- and C-terminal extensions as well as the insertions (dashed curves) between the secondary structure elements in the α/β barrel. Loops 1–3 (dark green, red, blue) have special names in the RNR literature for their involvement in specificity regulation. The gray secondary structure elements (starred) are partially integrated in the α/β barrel and are involved in substrate binding. (**B**) The barrel portion (top) of the *Escherichia coli* class Ia catalytic subunit (bottom dimer). The ATP-cone domain is colored in orange. (**C**) The barrel portion (top) of the *Thermotoga maritima* class II catalytic subunit (bottom dimer). (**D**) The barrel portion (top) of the T4 phage class III catalytic subunit (bottom dimer). In class III RNRs, the loop 1 region (dark green) is a long helix that is involved in dimerization.

The online version of this article includes the following figure supplement(s) for figure 1:

**Figure supplement 1.** Ribonucleotide reductases (RNRs) catalyze the reduction of ribonucleoside di- or tri-phosphates (NDPs or NTPs) to their respective deoxynucleotide forms (dNDPs or dNTPs).

Except for a subset of class II RNRs which are monomeric, all extant RNR α subunits are thought to dimerize at the αA and αB helices in their active forms. Class I and II RNRs dimerize in anti-parallel orientation at these helices, such that the dimer interface is capped on both ends with a functional insertion known as loop 1 (*Figure 1B–C*, dark green), which is the binding site for nucleotides that control the identity of the substrate in the active site (*Larsson et al., 2004*; *Zimanyi et al., 2016*). The binding of these so-called specificity-regulating effectors is coupled to substrate binding via another insertion known as loop 2 (*Figure 1B–C*, red) (*Zimanyi et al., 2016*). Class I and II RNRs also share another insertion, a long β-hairpin motif, between the αH helix and βH strand. Although less is known about the function of this insertion, it is thought to be important for binding the radical cofactor (the entire ferritin subunit in the case of class I RNRs and the AdoCbl cofactor in the case of class II RNRs)

(*Kang et al., 2020*; *Sintchak et al., 2002*; *Thomas et al., 2019*). Class III RNRs, on the other hand, dimerize at the αA and αB helices such that they are oriented parallel to each other across the interface. This orientation is stabilized by a long helix in place of the loop 1 insertion (*Logan et al., 1999*; *Figure 1D*, dark green). Additionally, the class III RNRs contain a glycyl radical domain with a zinc finger motif at the C-terminus. In contrast, the C-termini of class I and II RNRs that have been characterized are unstructured and contain cysteines to re-reduce the active-site cysteines via disulfide exchange (*Booker et al., 1994*; *Mao et al., 1992*; *Thomas et al., 2019*). Thus, overall, class I and II RNRs share more structural similarities with each other than they do with class III RNRs.

The ancestor of modern RNRs, and its subsequent evolution into the three known classes, has long been of interest for its hypothesized importance in transitioning life from an RNA/protein world to a DNA world (*Lundin et al., 2015*; *Lundin et al., 2009*; *Poole et al., 2002*; *Reichard, 1993*; *Stubbe, 2000*; *Stubbe et al., 2001*; *Torrents, 2014*; *Torrents et al., 2002*). Class I RNRs have been proposed to be the most recently evolved class as they require molecular oxygen, which became abundant as the Earth's atmosphere transitioned from anoxic to oxic. In contrast, class III RNRs have been proposed to have emerged before this transition as glycyl radicals and [4Fe-4S] clusters are both extremely oxygen-sensitive (*Fontecave et al., 1989*) and Fe-S chemistry is thought to have a prebiotic origin (*Goldman and Kaçar, 2021*). The AdoCbl chemistry used by class II RNRs, on the other hand, is neither oxygen-dependent nor especially oxygen-sensitive. Based on the relative oxygen sensitivities and biosynthetic complexities of the different cofactors, class III RNRs have been proposed to be the most ancient of the RNR family (*Reichard, 1997*; *Riera et al., 1997*). However, whether class III predates class II has been debated when considering cofactor availability (*Stubbe, 2000*). Indeed, cobalamin, AdoMet, and Fe-S clusters have all been proposed to be part of the cofactor set used by the last universal common ancestor (*Weiss et al., 2016*). It has been proposed in an alternative model that dimeric class II and III RNRs evolved in parallel from a shared ancestor using class II-like chemistry and that class I RNRs then evolved from class II (*Lundin et al., 2015*). However, this model was derived from a structure-based phylogenetic network (*Bryant and Moulton, 2004*), which estimates a phylogeny based on pairwise structural alignment scores rather than evolutionary modelling on protein sequences with well-established amino acid substitution models and statistical frameworks. Despite much effort to understand the diversity and evolution of RNRs, the reconstruction of a large-scale, unifying phylogenetic inference that includes all classes of RNRs has been hindered by the significant computational demand imposed by the high sequence diversity of the extant members of the family.

In this study, we used the conserved 10-stranded α/β barrel to anchor highly diverged RNR sequences in a structure-based alignment and used maximum-likelihood (ML) inference to reconstruct the largest RNR phylogeny to-date, consisting of 6779 α sequences, that unifies all classes. Our comprehensive phylogeny shows the parallel development of three major clades, corresponding to the three known classes, with a small, phylogenetically distinct clade, which we denote as class Ø (for convenience, pronounced 'oh'), placed as an ancestral clade to the class I and II RNRs. Using small-angle X-ray scattering (SAXS), cryogenic-electron microscopy (cryo-EM), and AlphaFold2 (*Jumper et al., 2021*), we show that the class Ø α is the most minimal RNR structurally characterized thus far. These observations were corroborated by evo-velocity analysis, an alignment-independent method that employs protein language models to infer sequence fitness and evolutionary trajectories (*Hie et al., 2022*). Together, our analyses indicate that class III RNRs diverged early on and evolved independently of class I and II RNRs, which diverged from an ancestor shared with the minimal, class Ø RNRs. Our evolutionary model provides an explanation for the structural similarities of the class I and II RNRs and also supports the idea that adaptations to an oxygenated atmosphere appeared earlier than initially thought.

## Results
### Phylogenetic reconstruction and evo-velocity analysis reveal the evolution of three RNR classes and a novel clade

To study the molecular evolution of the RNR family, we performed comprehensive phylogenetic inference on the catalytic α subunits (*Figure 2*). Unlike sequence analyses, such as sequence similarity network (SSN) analysis (*Atkinson et al., 2009*), which provide insight exclusively on the global

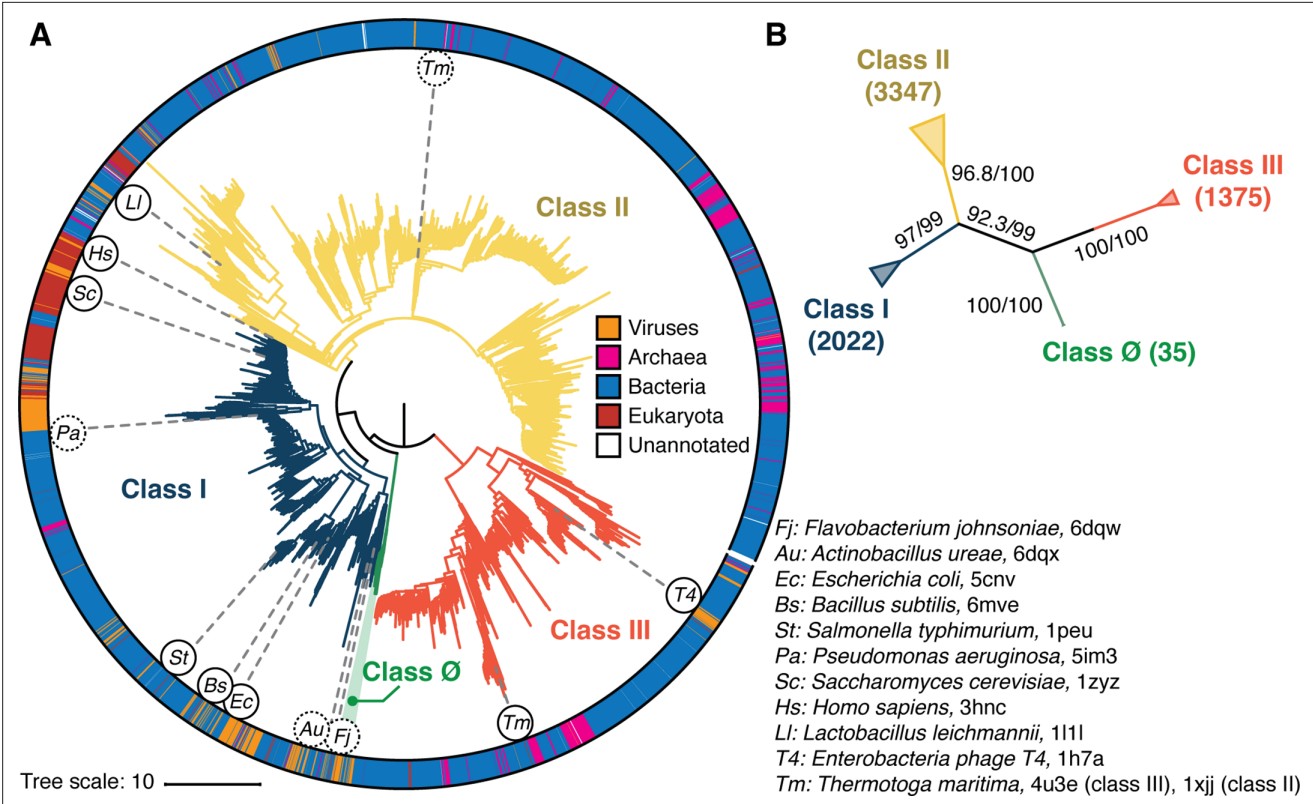

**Figure 2.** Phylogenetic reconstruction of the ribonucleotide reductase (RNR) family. (**A**) A representative phylogeny of 6779 extant RNR α sequences rooted at the midpoint. The superfamily forms four distinct lineages, with the three major clades (blue, yellow, red) corresponding to the three biochemically known classes I–III. Class III is the most distantly related clade (red). A small ancestral clade (green) forms an outgroup to the clades corresponding to classes I and II (blue and yellow, respectively). The scale bar represents the expected number of amino acid substitutions per site in the alignment. The taxonomic group each sequence belongs to is labeled by a color strip at the circumference. Sequences used for the structural alignment are mapped onto the tree by organism ID in circles. Organism IDs in dashed circles are α sequences that were utilized in the structural alignment but were filtered in favor of more representative sequences for the tree inference and thus are not on the tree (see Methods). These structures are represented by closely related sequences with ≥80% sequence identity. (**B**) An unrooted representation of the RNR phylogeny with branch supports (UFboot/SH-ALRT statistics) for the placements of major lineages. Deep nodes in the RNR phylogeny are resolved with high confidence.

The online version of this article includes the following figure supplement(s) for figure 2:

**Figure supplement 1.** All 20 inferred phylogenies of the ribonucleotide reductase family shown in the unrooted representation.

**Figure supplement 2.** Tree inference is robust to sequence identity threshold.

similarities shared between extant sequences, phylogenetic inference delineates the evolutionary history of related sequences utilizing complex models of evolution that account for the rate of mutation of base pairs or amino acids. However, large-scale phylogenetic reconstruction on entire protein superfamilies is technically challenging and computationally demanding. Homology is often difficult to detect between extensively diverged proteins, complicating the generation of a robust sequence alignment, without which accurate topological reconstruction and model parameterization cannot be achieved. To overcome these challenges, we adapted a recently developed workflow (*Spence et al., 2021*), which uses ensembles of hidden Markov models (HMMs) guided by structural information to build an accurate alignment of 6779 α sequences that spans five PFAM families.

Phylogenetic reconstruction was performed by ML inference. Tree-searches were performed with 10 replicates under the sequence evolution model LG+R10 (*Kalyaanamoorthy et al., 2017*; *Le and Gascuel, 2008*), which was selected by Akaike and Bayesian information criteria (*Dridi and Hadzagic, 2019*) on a representative subset of the full 6779 RNR sequence dataset. To test the robustness of our phylogenetic hypotheses against model diversity, we inferred an additional 10 replicates of tree-search under an alternative general amino acid replacement matrix (WAG+R10) (*Whelan and Goldman, 2001*). Each of the 10 phylogenies inferred under LG+R10 failed rejection by the approximately

unbiased (AU) test (*Shimodaira, 2002*) conducted to 10,000 replicates (lowest p-value = 0.283), including those whose branch supports failed to converge. All 20 inferences, irrespective of evolutionary model, converged on a similar phylogenetic topology. To further test the robustness of our results, we repeated our analyses at a redundancy threshold of 55% sequence identity, which yielded a topology with the same cladistic groupings as the full inference (*Figure 2—figure supplements 1 and 2*), indicating that the likelihood surface of the evolutionary relationships of RNR α subunits has a well-defined global maximum that we had reached in our tree-searches. This was corroborated by high branch supports (ultrafast bootstrap approximation 2.0 and approximate likelihood ratio tests) for major bifurcations in each independent reconstruction, including deep nodes at the midpoint root of each tree (*Figure 2B*). The topology that is presented in *Figure 2* had the highest branch supports at key nodes of interest in RNR evolution.

Consistent with our current understanding of RNR evolution, all reconstructed topologies that converged and failed rejection by the AU-test resolved the three biochemical classes of RNRs as monophyletic lineages (*Figure 2*, blue/yellow/red). Additionally, we find strong support for classes I and II diverging from a single common ancestor that shared an ancestor with the class III RNRs. We identified a novel clade of α sequences that diverged from the last common ancestor (LCA) of classes I and II, which we denote as the class Ø clade (*Figure 2*, green). We note that there are two trees inferred from the LG+R10 dataset that do not resolve class Ø as a separate clade, where instead these sequences are a monophyletic clade diverging from class II sequences. Topology testing shows that although these trees are valid hypotheses, the Δ log likelihood values indicate that they are the least likely trees of the 10 trees inferred under the LG+R10 model. Moreover, when we repeated phylogenetic reconstruction with a different sequence redundancy threshold, the topology with class Ø as an ancestral clade to classes I and II was reproduced independently (*Figure 2—figure supplement 2*). In the topology with the best statistical support, the branch that separates the LCA of classes Ø, I, II from class III is the midpoint of the phylogeny. Rooting on this branch places class III as the most ancestral, followed by the emergence of the LCA of classes Ø, I, II and the subsequent divergence of classes I and II. Midpoint rooting is the best available method to understand the diversification of all RNRs in this instance, given that the tree unifies all classes of RNRs in the in-group and is well balanced despite the diversity of sequences. The consistency in the location of the midpoint in the full-dataset phylogeny and the reduced redundancy phylogeny (*Figure 2—figure supplements 1 and 2*) further validates the choice to root on the midpoint.

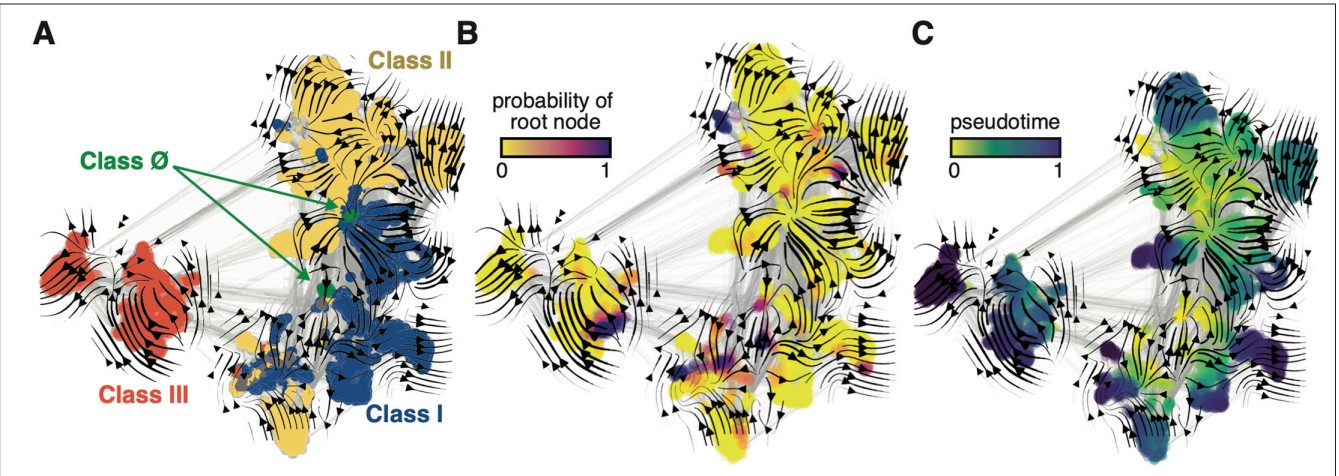

**Figure 3.** Evo-velocity analysis of the ribonucleotide reductase (RNR) family. ESM-1b embedded RNR sequences were projected onto a two-dimensional vector plot, where the horizontal and vertical axes are uniform manifold approximation and projection (UMAP) 1 and 2, respectively. Each colored point in the plot corresponds to one of the 6779 α sequences from the full RNR phylogeny. (**A**) Vector field plot colored by RNR classification: clade Ø (green), class I (blue), class II (yellow), class III (red). (**B**) Vector field plot colored by the probability of a sequence being a root of the sequence space, where purple sequences are the most likely to represent probable roots. (**C**) Vector field plot colored by pseudotime, a proxy for phylogenetic depth. Yellow (pseudotime = 0) represents ancestral sequences and indigo (pseudotime = 1) represents sequences that have diverged the most from ancestral sequences.

As phylogenetic inference is susceptible to biases introduced by taxon sampling (*Pollock et al., 2002*), model parameterization and assumptions, sequence alignment (*Simmons et al., 2011*), and tree-search and because our choice to midpoint root the phylogeny assumes a constant evolutionary rate, we also used a recently described phylogeny-independent method known as evo-velocity analysis to test our phylogenetic hypotheses (*Hie et al., 2022*). Evo-velocity is a machine-learning analysis based on the principles of natural language processing (NLP) (*Hie et al., 2022*). NLP models such as ESM-1b, a >650,000,000 parameter NLP model trained on UniProt50 amino acid sequences (*Rives et al., 2021*), can embed query sequences in a high-dimensional space where likelihoods can be numerically computed as proxy fitness scores on a graph network. By assuming that the inherent directionality of the embedded sequence graph network moves in the direction of evolution, as evolution compels proteins toward more fit (i.e., more probable in the NLP model) sequences, sequences with low learned probabilities can be interpreted as starting points in evolutionary trajectories. Evo-velocity thus bypasses many of the assumptions of phylogenetic inference as it is alignment independent, does not require model parameterization or tree-search, and has previously been used to validate conclusions based on superfamily-scale phylogenetic inference (*Hie et al., 2022*; *Spence et al., 2021*).

When projected onto a two-dimensional basis (see Methods), the ESM-1b embedded RNR graph network (*Figure 3*) shares many common topological features with the full family phylogeny (*Figure 2*). Class III RNRs (*Figure 3A*, red) belong to a diverged cluster of sequences that have evolved independently of class I and II sequences (*Figure 3A*, blue/yellow). Conversely, classes I and II are joined over the same sequence space, indicating a common recent ancestor. The class Ø sequences occupy the interface between classes I and II (*Figure 3A*, green), consistent with their phylogenetic placement as ancestral to the class I/II lineage. Evo-velocity analysis also identified multiple roots in the RNR family (*Figure 3B*, dark purple). This is likely a consequence of sequences diverging over geological timescales and lacking intermediate phylogenetic information between major groups. The multiple roots belong to each of the major lineages, signifying independent evolution down distinct trajectories. However, pseudotime velocity, a proxy for phylogenetic depth (*Haghverdi et al., 2016*), clearly captures the interface of class I and II sequences (including class Ø) as the most ancestral within that lineage (*Figure 3C*). The congruence between phylogenetic and evo-velocity analyses supports our hypotheses on the evolution of extant RNRs. The projection of class Ø sequences onto the ancestral interface of classes I and II additionally provides evidence that they indeed hold a pivotal position in RNR evolution.

## Class Ø RNRs are predominantly found in marine bacteria and cyanophages

The class Ø clade consists of RNRs from both bacteria and phages, many of which are phototrophs. Nearly half of the sequences in the clade were obtained from cyanophages (*Figure 4A*, bolded species) or uncultured phages, while the remainder were obtained from marine bacteria, including those collected in the *Tara* Oceans Expedition (*Tully et al., 2018*). Performing a BLAST search of photosynthetic genes, such as *psbA* and *psbD*, against the genomes (of which many are incomplete) of the 35 bacteria and phages listed in the clade yielded 11 hits (*Figure 4A*, species in green).

The classification of the cyanophage sequences observed in this clade (*Figure 4A*, bolded species) has been previously debated (*Harrison et al., 2019*). These RNRs from *cyanosipho-* and *cyanopodoviruses* were initially annotated as class II, based on studies of the *Prochlorococcus* phage P-SSP7 genome, which contains two consecutive genes that were proposed to represent a split RNR with homology to class II RNRs (*Dwivedi et al., 2013*; *Lindell et al., 2005*; *Sullivan et al., 2005*). More recently, it was discovered that these two genes encode for an RNR α subunit and a small ferritin-like protein and were thus reannotated as class I (*Harrison et al., 2019*). However, in the phylogenetic analysis reported in this previous study, the cyanophage α sequences fell in either the class I or class II clades depending on the stringency for homology used for grouping the sequences. This result contrasts with our work, where the class Ø clade is phylogenetically distinct from the other major clades (*Figure 2*). Additionally, we find that all completely sequenced operons with a class Ø RNR α gene (24/35 in *Figure 4A*) contain a small ferritin-like gene (annotated with InterPro family number for the ferritin-like superfamily IPR009078) immediately downstream (*Figure 4—figure supplement 1*), and thus, we find that this feature is not exclusive to cyanophages. Interestingly, an all-vs.-all

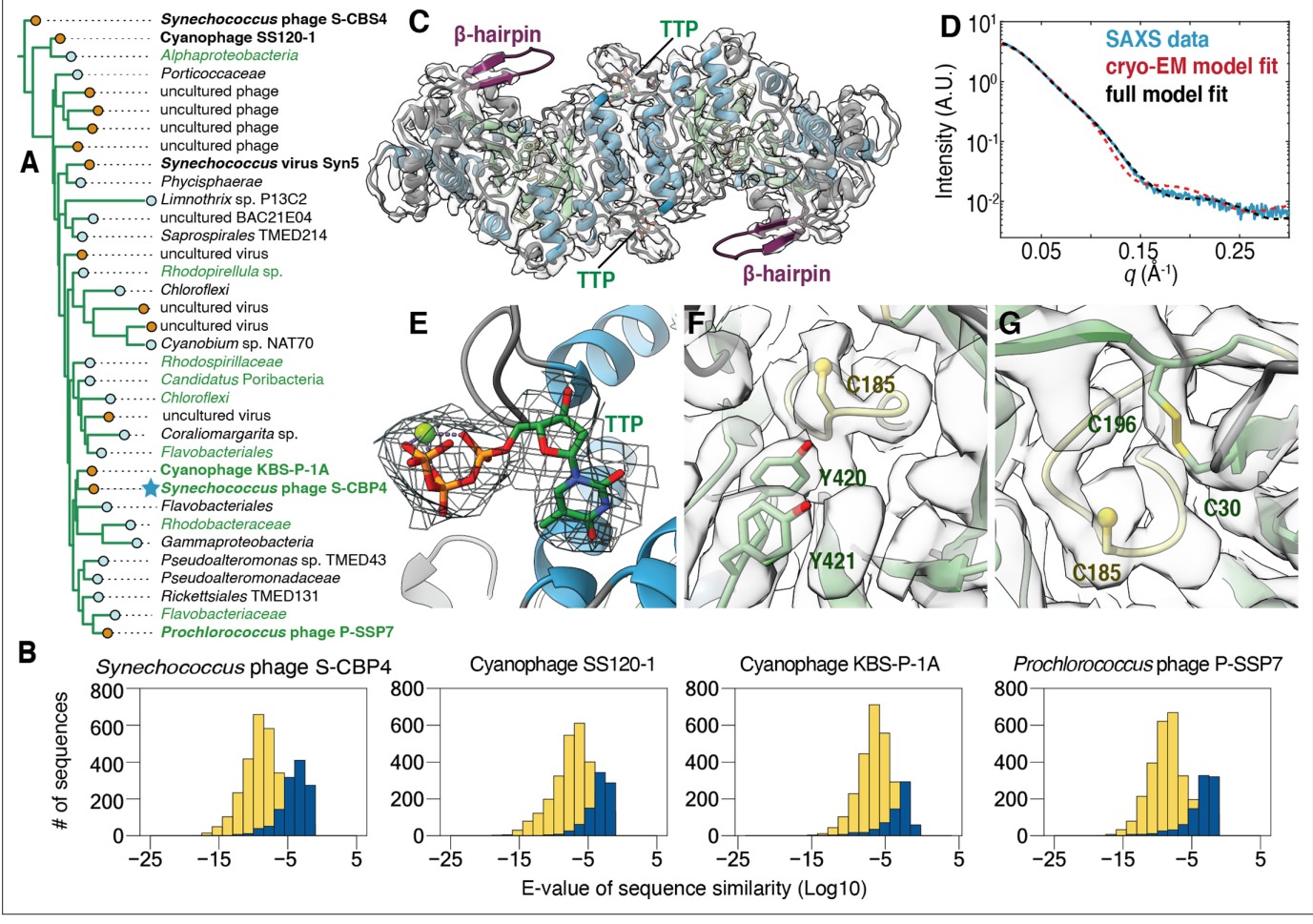

**Figure 4.** The class Ø α subunit shares similarities with both class I and II α subunits. (**A**) An expansion of the class Ø clade from the full tree in *Figure 2*. Cyanophage sequences in this clade are bolded. Genomes with identified photosynthetic genes, *psbA* and *psbD*, are colored in green. (**B**) Representative results from an all-vs.-all pBLAST search of every class Ø sequence against all ribonucleotide reductase (RNR) α sequences from other clades. Blue bars represent number of significant hits (E-values<10⁻³) of the title sequence to sequences in the class I clade. Yellow bars represent number of sequences from the class II clade. Overall, the class Ø clade shares greater homology with the class II RNRs. (**C**) A 3.46 Å cryogenic-electron microscopy (cryo-EM) map of the *Synechococcus* phage S-CBP4 α subunit (shown at a threshold of 2.17) depicts a dimer with thymidine triphosphate (TTP) bound at the allosteric specificity sites. The β-hairpin (violet) is a shared trait of class I and II RNRs. (**D**) Experimental small-angle X-ray scattering (SAXS) profile (blue solid) of the *Synechococcus* phage S-CBP4 α subunit in the presence of 200 µM TTP, 200 mM guanosine diphosphate (GDP) is explained well by the theoretical scattering of our cryo-EM model (red dashed). Model-data agreement is further improved by modeling the disordered N- and C-termini in AllosModFoXS (black dashed, see Methods) (***Schneidman-Duhovny et al., 2010***). Cryo-EM density for (**E**) TTP at the specificity site, (**F**) a stacked tyrosine dyad adjacent to the catalytic cysteine, and (**G**) the oxidized cysteine pair in the active site is shown at a threshold of 2.77.

The online version of this article includes the following figure supplement(s) for figure 4:

**Figure supplement 1.** Six representative operons of class Ø sequences.

**Figure supplement 2.** *Synechococcus* phage S-CBP4 α is a stable dimer in the presence of nucleotides.

**Figure supplement 3.** Cryogenic-electron microscopy (Cryo-EM) image processing workflow for reconstructing the 102 kDa *Synechococcus* phage S-CBP4 α₂ dimer.

**Figure supplement 4.** The class Ø α subunit features the most minimal ribonucleotide reductase (RNR) architecture discovered to-date.

**Figure supplement 5.** One structural difference between class I and II α subunits is the packing of βJ residues next to the catalytic cysteine on the finger loop (yellow).

**Figure supplement 6.** Comparison of class Ø ferritin-like proteins with class I ribonucleotide reductase (RNR) β subunits.

pBLAST search of every class Ø sequence against all class I sequences in our phylogeny detects poor homology (E-value > 0.05) (*Figure 4B*, blue), while querying class Ø RNRs against class II returns hundreds of significant hits (E-value < 1E-10) (*Figure 4B*, yellow). Thus, by sequence homology, class Ø RNRs appear to be more class II like despite being associated with a ferritin-like gene that is reminiscent of class I RNRs, which supports the phylogenetic results indicting they are a distinct class.

## Class Ø RNRs are minimal and share features of class I and II enzymes

The class Ø lineage consists of α sequences with an average length of 412 amino acids, which are significantly shorter than those of the class Id catalytic subunit that were previously described as having a minimal architecture with a median length of 565 amino acids (*Rose et al., 2019*). Like the class Id α sequences, the class Ø sequences all lack an ~100-residue ATP-cone motif at the N-terminus, which is used for allosteric regulation of overall activity in many other RNRs (*Meisburger et al., 2017*). Additionally, the class Ø C-terminus contains a double cysteine motif, which is likely used for reducing a pair of active-site cysteines that are oxidized following every turnover. Of the 24 class Ø sequences that are complete at the C-terminus, 10 contain a CXXC motif and 14 contain a CXC motif (where aspartic acid is the most common middle amino acid). The sequences that appear to be incomplete at the C-terminus largely belong to uncultured phages. The C-terminal CXXC motif is a common feature of class I RNRs (found in 1737 of 2022 class I sequences in our dataset), but the CXC motif is rare among both class I and II RNRs. Only seven class I sequences and one class II sequence in our dataset appear to have a CXC motif. The prevalence of this motif makes the class Ø RNRs unlike class I or II RNRs. Interestingly, the CXXC-containing sequences are most ancestral within the class Ø clade, while the more recently evolved sequences have the CXC motif.

To examine the structural features of the class Ø RNR, we used a combination of AlphaFold2, SAXS, and cryo-EM to characterize the *Synechococcus* phage S-CBP4 α subunit. Importantly, this α sequence is highly representative of the clade, scoring an E-value of 1.2E-228 on an HMM sequence profile of each class Ø sequence. In the absence of nucleotides, our SAXS profiles are consistent with a monomer-dimer equilibrium (*Figure 4—figure supplement 2*) where the S-CBP4 α subunit is predominantly monomeric at low protein concentrations (≤4 µM). Addition of substrate (guanosine diphosphate [GDP]) does not shift this equilibrium, but the presence of the corresponding specificity effector (thymidine triphosphate [TTP]), strongly favors dimerization (*Figure 4—figure supplement 2A, B*). This result is consistent with TTP binding the specificity site to stabilize the same dimeric form adopted by class I and II α subunits. Using solution conditions that saturate nucleotide binding (200 µM TTP, 200 µM GDP), we obtained a cryo-EM map of the S-CBP4 $\alpha_2$ dimer to 3.46 Å resolution (Fourier shell correlation [FSC] = 0.143) (*Figure 4C*, *Figure 4—figure supplement 3*). An atomic model (residues 22–426 out of 470) was refined against the cryo-EM map using an AlphaFold2 prediction as the starting model (*Table 1*). The final refined model was highly similar to the starting model with a root mean square deviation (all-atom RMSD) of 1.44 Å. The elongated shape of the S-CBP4 $\alpha_2$ dimer observed by cryo-EM (*Figure 4D*, red dashed curve) explains the distinct flattened region in our experimental SAXS profile between q~0.05–0.15 Å$^{-1}$ (*Figure 4D*, blue curve), and excellent model-data agreement is observed over the full q-range when the disordered N- and C-termini are included in the model (*Figure 4D*, black dashed curve and blue curve).

Overall, the S-CBP4 α subunit displays the most minimal RNR architecture discovered to-date with few insertions about the catalytic barrel (*Figure 4—figure supplement 4*). The minimal set of insertions includes traits shared by class I and II RNRs, such as loop 1, where we find cryo-EM density for bound TTP (*Figure 4E*). Additionally, the S-CBP4 α subunit contains a long β-hairpin following the βH strand of the catalytic barrel that in class II RNRs is thought to be important for cofactor binding (*Sintchak et al., 2002*) and in class I RNRs is thought to be important for interactions with the β subunit (*Kang et al., 2020*). Although we do not observe density for the C-terminus past the βJ strand of the catalytic barrel, two consecutive tyrosine residues (Y420 and Y421) on βJ are well resolved and stacked adjacent to the catalytic cysteine (C185) at the tip of finger loop (*Figure 4F*). In class I RNRs, this stacked arrangement of the tyrosine dyad is required for long-range proton-coupled electron transfer (PCET) between the β subunit and the catalytic cysteine (*Greene et al., 2017*; *Figure 4—figure supplement 5A*). By comparison, inspection of existing structures (*Larsson et al., 2010*; *Sintchak et al., 2002*) suggests that in class II RNRs, the space between the βJ strand and the finger-loop cysteine forms a pocket to accommodate the adenosyl group of the cobalamin cofactor

**Table 1.** EM data collection, processing, and refinement.

Data collection and processing

| | |
|---|---|
| Microscope | Talos Arctica |
| Camera | K3 |
| Magnification | 79,000 |
| Voltage (keV) | 200 |
| Electron exposure (e$^-$ Å$^{-2}$) | 50 |
| Defocus range (µm) | −0.6 to −2.0 |
| Pixel size (Å) | 1.07 |
| Micrographs used (no.) | 432 |
| Initial particles (no.) | 581,884 |
| Final particles (no.) | 107,885 |
| Symmetry imposed | C2 |
| Map resolution (Å)<br>FSC threshold | 3.46<br>(0.143) |
| Map resolution range (Å) | 3.0–7.0 (75%) |

Refinement

| | |
|---|---|
| Initial model used | AlphaFold2 prediction for UniProt entry M1PRZ0 |
| Model resolution (Å)<br>FSC threshold | 3.6<br>(0.5) |
| Map sharpening B factor (Å$^2$) | −171 |
| Model composition | |
| Non-hydrogen atoms | 6260 |
| Protein residues | 802 |
| Ligands | TTP |
| B factors (Å$^2$) | |
| Protein | 74.37 |
| Ligand | 67.23 |
| r.m.s. deviations | |
| Bond lengths (Å) | 0.004 |
| Bond angles (°) | 0.937 |
| Validation | |
| MolProbity score | 1.47 |
| Clashscore | 4.69 |
| Poor rotamers (%) | 0.00 |
| Ramachandran plot | |
| Favored (%) | 96.49 |
| Allowed (%) | 3.51 |
| Disallowed (%) | 0.00 |

(*Figure 4—figure supplement 5B*). We note that the appearance of the double tyrosine motif does not necessarily indicate that an RNR α sequence belongs to class I. In fact, there are class II sequences with this motif, but structure prediction of these sequences indicates that the double tyrosine motif is placed outside of the binding pocket for adenosyl group, rather than filling this space (*Figure 4— figure supplement 5C*). Finally, we observe a disulfide between active-site cysteines (C30 and C196) that in class I and II RNRs serve as reducing equivalents during nucleotide reduction (*Figure 4G*). Consistent with an oxidized active site, cryo-EM density for the substrate is not observed. This in turn explains why residues 13–20 are disordered in our cryo-EM model, as it is predicted to form a helix involved in substrate binding (*Figure 1A*, gray helix adjacent to left star).

Although the class Ø α subunit contains a tyrosine dyad poised for PCET like class I α subunits, the class Ø ferritin-like sequences are significantly shorter (average length of 240 amino acids) than those of bona fide class I β subunits (average length of 346 amino acids). Querying each complete class Ø ferritin-like sequence returns only a single homologous sequence over the whole length of the alignment (all hits E-value > 1E-4, excluding one) when searched against PFAM families PF00210 (ferritin-like proteins, 17,981 entries) and PF00268 (class I RNR β subunit, 9898 entries). However, based on AlphaFold2, which detects more than sequence homology, class Ø ferritin-like proteins resemble class I β subunits with shorter insertions and termini (*Figure 4—figure supplement 6A-D*). As in class I β subunits, the class Ø proteins are predicted to contain a core ferritin fold consisting of two helix-turn-helices (*Figure 4—figure supplement 6A and C*, red/pink and yellow/green), each contributing an E/D+EXXH metal-binding motif (*Ruskoski and Boal, 2021*). Additionally, while most class I β sequences have a tyrosine serving as the site of the radical downstream of the first metal-binding motif (*Figure 1—figure supplement 1*), class Ø ferritin-like proteins contain a redox-inert residue (Phe or Leu) at this site, much like class Ic RNRs (*Bollinger et al., 2008*; *Cotruvo and Stubbe, 2011*; *Figure 4—figure supplement 6E*). Class Ø ferritin-like sequences also contain a C-terminal tyrosine (Y239 in *Figure 4—figure supplement 6A*), which in class I RNRs is involved in inter-subunit radical transfer (Y356 in *Figure 4—figure supplement 6C*). However, the C-termini of the class Ø ferritin-like proteins are unusually short, terminating at the residue downstream from this tyrosine, and lack the ~10-residue extension that class I β subunits use for binding to the α subunit. Thus, although the class Ø ferritin-like proteins are likely to use a similar radical-generating mechanism as class I β subunits, they may interact with the α subunit in a different manner to enable PCET. However, ongoing biochemical work to confirm this has thus far been non-trivial as the class Ø ferritin-like proteins can be unstable in the absence of metals, most likely because of their small size. This makes characterization of metal dependence more challenging as it means that the metal composition cannot be easily varied. Further investigation will be the focus of future work. Regardless, the class Ø α structure illustrates how the class I and II RNRs may have inherited shared traits from a common ancestor.

## Discussion

As RNR sequences have been diverging for billions of years, pairwise sequence identity is frequently low (<15%), and sequence homology is often undetectable between distant members of the enzyme family. It is therefore challenging to produce an accurate sequence alignment of all RNR classes, without which a phylogenetic inference cannot be performed. Thus, although various models for RNR evolution have been proposed by biochemical reasoning (e.g., based on the availabilities and oxygen sensitivities of the cofactors), prior to our work, phylogenetic inference had only been performed on a very small scale. Early studies included an inference of eight class II sequences (*Jordan et al., 1997*) performed with Clustal W (*Thompson et al., 1994*) and an inference with a dataset of 96 class I, II, and III sequences (*Torrents et al., 2002*). In the latter study, Torrents and co-workers used a neighbor-joining (NJ) algorithm to build a tree using a Poisson model, which assumes that all amino acids have equal probability of change (*Torrents et al., 2002*). The NJ approach can be understood as a way to build a tree by creating nodes between pairs of sequences that are considered close by a certain metric for distance, which in this case is determined by the Poisson sequence evolution model. This approach differs significantly from modern methods of tree inference in which the tree topology is determined by the sequence evolution model itself (*Minh et al., 2020*) often in replicate. Furthermore, the sequence availability at the time was too sparse to capture the true diversity of the RNR family, and the dataset lacked class Ø sequences, monomeric class II sequences, as well as many subclasses of the class I sequences.

Since then, small-scale inferences have been performed on subsets of the RNR family on either the α or β subunit with datasets ranging between 10s and 100s of sequences in size (*Dwivedi et al., 2013*; *Lundin et al., 2010*; *Martínez-Carranza et al., 2020*; *Rose et al., 2019*; *Rozman-Grinberg et al., 2018*; *Sakowski et al., 2014*). Among these, the work of Lundin and co-workers has been the most comprehensive with respect to consideration of the different RNR classes (*Lundin et al., 2010*). The growth of sequence information, however, also meant that greater diversity had to be accounted for by phylogenetic methods, and therefore, phylogenetic inference had not been performed on all RNR classes simultaneously since the seminal work of Torrents et al. In its place, a structure-based phylogenetic network was constructed via an NJ algorithm, where the distance between two sequences was estimated by pairwise structural alignment scores (*Lundin et al., 2015*). Under the assumption that this network estimates a phylogenetic tree, different possible roots were considered, and although none led to monophyly of the RNR classes, it was proposed that the class I RNRs emerged from the dimeric class II RNRs.

In this study, we succeeded in performing a large-scale phylogenetic reconstruction of the entire, highly diverse RNR family for the first time by using structural information to aid the sequence alignment procedure. To test for alternative hypotheses, we performed ML inference using two different amino acid replacement matrices in replicate (10 trees for each model) and repeated the inference with a different sequence redundancy threshold to test for the effect of taxon sampling on tree topology and evolutionary conclusions. Key bifurcations featured high branch supports and reproducibility in replicate tree inferences, allowing us to make robust interpretations, which were further supported by evo-velocity analysis, a recently introduced method that is independent of multiple sequence alignments (*Hie et al., 2022*). In contrast to previously proposed models (*Lundin et al., 2015*), our unified RNR phylogeny supports the evolution of three major, monophyletic clades corresponding to the three biochemically known classes as well as a new phylogenetic clade, which we denoted as class Ø. Together, our bioinformatic analyses support an evolutionary trajectory in which the $O_2$-sensitive class III RNRs are the earliest to diverge, and the class Ø clade shares an ancestor with the LCA of the $O_2$-dependent class I and $O_2$-tolerant class II RNRs. It is important to note that all evolutionary models, including ours, are hypotheses, and it is impossible to know the true historical trajectory of mutations. Nonetheless, we can say that the evolutionary conclusions we present in this work were derived from the most advanced approach that is currently possible, and with the strongest statistical support and congruence across multiple independent methods. This work highlights the advances we have seen in bioinformatic methods while also providing a roadmap for the many important questions that remain.

Most notably, our phylogeny revealed the novel class Ø clade, which is exclusively (among characterized organisms) associated with marine microbes, such as cyanophages and photosynthetic bacteria. Bioinformatically, we showed that the class Ø α subunit shares significant sequence homology with class II sequences, while also containing in its gene neighborhood a gene for a ferritin-like protein that resembles the class I β subunit. Structurally, we further showed that the class Ø α subunit defines the minimum set of features shared by class I and II RNRs, including the orientation of the two monomers in the dimer, specificity site at loop 1, and the β-hairpin. It was previously hypothesized that the LCA of RNRs may have possessed class II-like biochemistry based on the Precambrian bioavailabilities of iron and cobalt and the abiotic synthesis of porphyrin rings (*Lundin et al., 2015*). An alternative hypothesis is supported by our phylogeny, which places the class I mechanism as ancestral to, or having diverged in parallel with, that of class II. Although we cannot classify the mechanism of radical generation in the class Ø RNRs in the absence of biochemical data, our phylogeny suggests that a class II-like enzyme resembling the extant class Ø sequences recruited a ferritin-like protein into the catalytic machinery, before diverging into class I RNRs (which retained a ferritin subunit for catalysis) and class II RNRs (which specialized in cobalamin usage). The usage of a ferritin-like protein in class Ø RNRs is supported by the observation of a stacked YY dyad in the active site of our cryo-EM structure, which suggests that long-range radical transfer is an important component of catalysis in these enzymes. However, the short C-terminus of the class Ø ferritins and the prevalence of the CXC motif in the α C-terminus suggest that there are key differences with the class I and II RNRs. Future biochemical characterization will require identification of the correct metallocofactors for activity and possible reducing partners.

Based on our results, we present the following model for the evolution of the RNR family. The LCA of the RNR catalytic subunit likely shared the most biochemical resemblance to extant class III

RNRs. Increasing oxygen availability selected for the divergence of the RNR family into the anaerobic LCA of class III and the oxygen-tolerant LCA of classes Ø–II, from which the minimal class Ø RNRs diverged. Class I and II RNRs sharing an ancestor would suggest that multiple strategies to adapt to the presence of oxygen evolved in parallel, rather than sequentially. Interestingly, we find sequences from the earliest diverging cyanobacterium, *Gloeobacter*, near the root of the class II clade in our phylogeny. *Gloeobacter* are notable for lacking thylakoid membranes and have been called 'the missing link' between anoxic photosynthesis, which involves Fe-S chemistry, and oxygenic photosynthesis (*Nakamura et al., 2003*; *Saw et al., 2013*). Thus, along with the emergence of cyanobacteria (*Schirrmeister et al., 2013*), it is possible that class II RNRs evolved well before the Great Oxygenation Event (GOE). If so, this would imply that class I RNRs also emerged before the Earth's atmosphere became permanently oxidizing. In fact, recent work suggests that oxygen availability was sufficient for the birth of $O_2$-producing and $O_2$-utilizing enzymes to occur well before the geochemically defined GOE (*Jabłońska and Tawfik, 2021*). Such an evolutionary model could explain how $O_2$-dependent class I RNRs can share an ancestor with class II RNRs. In this respect, it is especially intriguing that class Ø RNRs are associated with cyanophages and photosynthetic bacteria.

## Outlook

With a unifying RNR phylogeny, we propose an evolutionary model with implications for the early emergence of molecular adaptations to oxygen. Our phylogeny also directs new avenues of study. We expect biochemical studies of the class Ø clade to provide insight into how an ancestral radical-generating mechanism may have diverged into class I and II mechanisms. We also anticipate that by analyzing the extensions and insertions about the catalytic barrel, we will gain a greater understanding of how allosteric and catalytic mechanisms evolved within the RNR family. Finally, with a phylogenetic inference, it is conceivable that we will be able to reconstruct ancestral sequences and gain detailed insight into evolutionary trajectories. In all, our unifying RNR phylogeny will guide future studies to fill important gaps in our ever-evolving understanding of this family of complex enzymes.

## Methods

### Sequence collection and SSN analysis

A dataset of 105,904 sequences belonging to families PF02867, PF08343, PF13597, PF00317, PF17975, and PF08471 was downloaded from the PFAM server (*El-Gebali et al., 2019*). The computational complexity of phylogenetic reconstruction increases exponentially with the number of taxa sampled in the inference. A sequence redundancy threshold of 85% was imposed on the dataset, and redundant sequences were clustered and represented as a single sequence in CD-HIT (*Fu et al., 2012*) to both balance taxon sampling across divergent lineages and reduce the computational complexity of the inference. Because the PFAM database typically features many sequences that are truncated, incomplete or are part of larger multi-domain proteins, we removed all sequences that were smaller or larger than 400 or 1100 residues (the expected upper and lower bounds for RNR sequences based on prior data and characterized proteins), respectively. An all-vs.-all pBLAST search was performed for each sequence in the dataset using default search parameters and an E-value threshold of 1E-240. The search result was visualized as a force-directed graph network in Cytoscape (*Hie et al., 2021*; *Shannon et al., 2003*). The final non-redundant dataset consisted of 12,968 sequences.

### Sequence alignment

As RNR sequences have been diverging for billions of years, pairwise sequence identity is frequently low (<15%), and homology is often undetectable between distant members of the superfamily. Conventional alignment algorithms failed to align these large and diverse datasets, which were evaluated against structural alignments of phylogenetically diverse RNRs. Instead, we employed our recently reported workflow (*Spence et al., 2021*) that uses ensembles of HMMs that are iteratively trained on increasingly diverse sequences until the full diversity of the dataset is captured. The initial HMM in this workflow captures residues that are tolerated at each site of the consensus RNR fold. This HMM was built from a non-redundant structural alignment of nine diverse RNR crystal structures (1h78, 4coj, 3o0m, 6cgm, 1pem, 5im3, 3hne, 3s87, and 2x0x) that had been aligned in MUSTANG (*Konagurthu et al., 2010*). HMMs were built using the HMMBuild application of the HMMer suite

(*Finn et al., 2011*). Sequences from each SSN cluster were scored against a cluster-specific HMM, which was generated in HMMer from all sequences within that cluster. The E-value that a sequence scores against the HMM of the cluster that it belongs to therefore serves as a metric for how representative that sequence is to the overall cluster. The most representative sequence from each sequence cluster in the SSN was aligned to the initial structural HMM (755 sequences).

This 'representative' alignment, and a corresponding ML phylogenetic tree that was reconstructed in IQ-TREE (*Minh et al., 2020*) was used as a seed alignment and guide tree, respectively, for the full alignment of all non-redundant RNR sequences. The final sequence alignment was performed in UPP from the SEPP alignment package (*Nguyen et al., 2015*). As this alignment workflow exploits structural information, residues that are not crystallographically resolved or residues that do not align, such as non-conserved insertions and N/C-terminal extensions (including the ATP cone), are not included in the alignment. The full alignment was manually refined by removing non-conserved insertions (typically single residue insertions conserved in fewer than 1% of sequences) and poorly aligned sequences that are unlikely to be functional RNRs. The final alignment features 6779 RNR sequences.

## Phylogenetic reconstruction

The RNR superfamily phylogeny was reconstructed by ML from the full RNR alignment. Phylogenetic inference was performed using IQ-TREE 2.0 (*Minh et al., 2020*) on the Australian National Computing Infrastructure (NCI) GADI supercomputer and BioHPC servers at the Cornell Institute of Biotechnology. Stochastic tree-search was conducted with a stochastic perturbation strength of 0.2 to a maximum of 2000 iterations. Convergence was defined as 200 iterations of tree-search that failed to improve the current best tree. Branch supports were measured by ultrafast bootstrap approximation 2.0 (*Hoang et al., 2018*) and the approximate likelihood ratio test, each conducted to 10,000 replicates. The sequence evolution model, LG replacement matrix (*Le and Gascuel, 2008*) with a 10 category free-rate rate heterogeneity model (*Kalyaanamoorthy et al., 2017*), was determined by ML using ModelFinder (*Kalyaanamoorthy et al., 2017*) as implemented in IQ-TREE2.0 on the representative sequence alignment. To test the robustness of our hypotheses against model uncertainty, we performed an additional 10 replicates of ML tree-search using an alternate general amino acid replacement matrix (WAG+R10) (*Whelan and Goldman, 2001*). The AU-test conducted to 10,000 replicates was used to compare all tree topologies (*Shimodaira, 2002*). To further test the robustness of our hypotheses against taxon sampling, we performed an additional and independent phylogenetic reconstruction on a sequence dataset with redundancy reduced to 55% sequence identity. As with the full inference, the 55% redundancy phylogeny was inferred in IQ-TREE 2.0 under LG+R10.

## ESM-1b evo-velocity modeling

Protein NLP models can provide valuable and phylogeny-independent evolutionary insight (*Alley et al., 2019*; *Hie et al., 2021*; *Rives et al., 2021*). The 6779 sequences in our dataset were embedded as high-dimensional vectors in the ESM-1b protein language model (*Rives et al., 2021*) and projected onto a two-dimensional directed network graph with uniform manifold approximation and projection (*McInnes et al., 2018*). Likelihoods along each edge in the network graph were computed and visualized as a vector field. Under the assumption that likelihoods inferred from the NLP model are correlated with evolutionary fitness, the vector field 'flows' in the direction of evolution, and pseudotime velocity is correlated with phylogenetic depth (*Hie et al., 2021*).

## Expression and purification of *Synechococcus* phage S-CBP4 α subunit

The gene for the α subunit of *Synechococcus* phage S-CBP4 RNR was synthesized by Genscript with a $His_6$-SUMO (smt3) tag at the N-terminus and cloned into a pET-28c+ vector in-between the *NcoI* and *XhoI* cloning sites. $His_6$-smt3-tagged-α was grown overnight at 37°C with 200 rpm shaking in 100 mL of LB medium supplemented with 50 μg mL$^{-1}$ kanamycin. Large-scale growth was initiated by adding 10 mL of saturated starter culture to six 1 L volumes of kanamycin-supplemented LB medium incubating at 37°C with 200 rpm shaking. At an $OD_{600}$ of 0.4, IPTG was added to a final concentration of 0.4 mM, and cultures were shaken at 37°C 200 rpm for 6 hr. Cells were harvested by centrifugation at 4°C (15 min at 3500×*g*), frozen in liquid nitrogen, and stored at –80°C.

All steps of purification were completed at 4°C. The cell pellet was placed on ice and thawed for 30 min prior to resuspension in Buffer A: 50 mM sodium phosphate, pH 7.6, 300 mM NaCl, 2 mM

imidazole, 5% (v/v) glycerol, and 1 mM tris(2-carboxyethyl)phosphine (TCEP). For lysis, Buffer A was supplemented with 5 U mL$^{-1}$ DNase, 5 U mL$^{-1}$ RNase A and 0.2 mg mL$^{-1}$ lysozyme (hen egg white, Sigma), 1.5 mM MgCl$_2$, and EDTA-free protease inhibitor tablet (Roche), 250 μM phenylmethylsulfonyl fluoride. Cells were homogenized with a French Press at 14,000 psi for 10 min inside a cold room operating at 4°C. Cell debris was separated by centrifugation at 4°C (25,000×*g*, 30 min), and the supernatant was loaded onto a Talon cobalt affinity column (5 mL bed volume) equilibrated in Buffer A. The column was washed with the equilibration buffer for 10 column volumes, and protein was eluted with Buffer A with 500 mM imidazole. Protein-containing fractions were pooled and buffer exchanged into Buffer A via a centrifugal concentrator (30 kDa Amicon). Smt3-protease was added to a 350:1 mole ratio of tagged protein:Smt3 protease. Detagged protein was collected via the flowthrough of a second cobalt affinity column equilibrated with Buffer A. Further purification was performed with size exclusion chromatography on a HiLoad Superdex 200 pg preparative 16/600 column in Buffer A. All protein concentrations are given as monomer concentrations. All biophysical studies were performed in the following assay buffer except where noted in the cryo-EM sample preparation: 50 mM HEPES pH 7.6, 150 mM NaCl, 15 mM MgCl$_2$, 1 mM TCEP, and 5% (v/v) glycerol.

## Small-angle X-ray scattering

X-ray scattering experiments were performed at the Cornell High Energy Synchrotron Source (CHESS) ID7A station. Data were collected using a 250 μm × 200 μm X-ray beam with an energy of 9.9 keV and a flux of ~10$^{12}$ photons s$^{-1}$ mm$^{-2}$ at the sample position. SAXS images were collected on an Eiger 4 M detector covering a range of q=0.009–0.55 Å$^{-1}$. Here, the momentum transfer variable is defined as q=4π/$\lambda$ sinθ, where $\lambda$ is the X-ray wavelength and 2θ is the scattering angle. Data processing at the beamline was performed in BioXTAS RAW (*Hopkins et al., 2017*). Briefly, scattering images were integrated about the beam center and normalized by transmitted intensities measured on a photodiode beamstop. The integrated protein scattering profile, I(q), was produced by subtraction of background buffer scattering from the protein solution scattering. Radii of gyration (R$_g$) were estimated with Guinier analysis, and pair distance distribution analysis was performed with Bayesian indirect Fourier transformation (*Hansen, 2000*). Error bars associated with R$_g$ values are curve-fitting uncertainties from Guinier analysis. Subsequent analysis was performed in MATLAB and utilized REGALS and other established protocols (*Meisburger et al., 2021*; *Meisburger et al., 2016*; *Skou et al., 2014*).

The scattering of *Synechococcus* phage S-CBP4 α subunit was first measured over the concentration range of 4–40 μM. Eight μM was chosen for subsequent titration of nucleotides to maintain near physiological concentration (*Ando et al., 2011*; *Parker, 2017*). For all titration experiments, background subtraction was performed with carefully matched buffer solutions containing identical concentrations of nucleotides following established protocols (*Skou et al., 2014*). For each measurement, 40 μL of sample were prepared fresh and centrifuged at 14,000×*g* at 4°C for 10 min immediately before loading into an in-vacuum flow cell kept at 4°C. For each protein and buffer solution, 20×2 s exposures were taken with sample oscillation to limit radiation damage then averaged together to improve signal. Singular value decomposition (SVD) was performed in MATLAB.

Size exclusion chromatography-coupled SAXS (SEC-SAXS) experiments were performed using a Superdex 200 Increase 10/300 GL (24 mL) column operated by a GE Äkta Purifier at 4°C with the elution flowing directly into an in-vacuum X-ray sample cell. To account for an ~10-fold dilution of the sample during elution, 40 μL sample was prepared at 80 μM protein in assay buffer with 200 μM TTP and 200 μM GDP for substrate. Sample was then centrifuged at 14,000×*g* for 10 min at 4°C before loading onto a column pre-equilibrated in a matched buffer containing 200 μM TTP, 200 μM GDP. Samples were eluted at a flow rate of 0.05 mL min$^{-1}$ and 2 s exposures were collected throughout elution until the elution profile had returned to buffer baseline. Normalized, integrated scattering profiles were binned sixfold in frame number and fourfold in q, and scattering profiles of the elution buffer were averaged to produce a background-subtracted SEC–SAXS dataset. SVD was performed to determine the number of significant components, and the SEC-SAXS dataset was decomposed in the MATLAB implementation of REGALS (*Meisburger et al., 2021*).

Structural modeling was performed using the ATSAS package (*Manalastas-Cantos et al., 2021*) and AllosModFoXS *Schneidman-Duhovny et al., 2010*; *Weinkam et al., 2012* following previously established protocols (*Ando et al., 2011*; *Ando et al., 2016*; *Meisburger et al., 2016*; *Thomas et al., 2019*). Theoretical scattering curves of the cryo-EM model (Methods described below, PDB: 7urg)

were calculated in CRYSOL (*Svergun et al., 1995*) with 50 spherical harmonics, 256 points between 0 and 0.5 Å$^{-1}$, and the default electron density of water. The overall scale factor and solvation parameters were determined by fitting to the protein scattering curve extracted by REGALS. Disordered and missing residues were modeled in AllosModFoXS (*Konarev et al., 2003*; *Manalastas-Cantos et al., 2021*) with sampling of static structures consistent with the starting model.

## Cryo-EM grid preparation and data acquisition

QuantiFoil holey carbon R 1.2/1.3 300-mesh grids were glow discharged on a PELCO easiGlow system for 45 s with 15 mA current. Grid freezing was then performed on an FEI Vitrobot Mark IV with the chamber humidity set to 100% and the temperature set to 4°C. Three µL of sample (4 µM *Synechococcus* phage S-CBP4 α subunit in 50 mM HEPES pH 7.6, 150 mM NaCl, 7.55 mM MgCl$_2$, 200 µM TTP, 200 µM GDP, 1 mM TCEP, 1% v/v glycerol) was applied onto the grid. The sample was blotted for 4 s and then immediately plunged into liquid ethane cooled by liquid nitrogen.

Data collection was performed at the Cornell Center for Materials Research (CCMR) on a Talos Arctica (Thermo Fisher Scientific) operating at 200 keV with a Gatan K3 direct electron detector and BioQuantum energy filter at a nominal magnification of ×79,000 (1.07 Å pixel$^{-1}$). A total of 856 movies was collected with a nominal defocus range from –0.6 to –2.0 µm and a total dose of 50 e$^-$ Å$^{-2}$ over 50 frames (2.164 s total exposure time, 0.0435 s frame time, 26.96 e$^-$ Å$^{-2}$ s$^{-1}$ dose rate).

## Cryo-EM data processing

Initial processing was performed in cryoSPARC v3.3.1 (*Punjani et al., 2017*). Patch motion correction and patch CTF estimation were performed on 856 movies. The resulting micrographs were manually curated based on statistics and visual inspection, and 432 micrographs were retained. Forty-six high-quality micrographs were then selected, from which the blob picker routine was used to pick particles. The resulting 99k particles were extracted and subjected to 2D classification, and the top four unique 2D classes were selected and used as templates for template picking on the entire dataset. Due to the large variance in ice conditions in many of our micrographs, masks were manually defined for every micrograph, and particle picks outside the ideal ice region were excluded. The resulting 582k particles were extracted with a box size of 256 pixels binned to 128 pixels. Two rounds of 2D classification were performed, and only particles corresponding to the top 2D classes with secondary structure features were kept. The 203k remaining particles were re-extracted with 256-pixel box size and subjected to ab initio reconstruction and heterogeneous refinement into two classes. The top class containing 117k particles was then subjected to homogeneous refinement with C2 symmetry, which yielded a 4.04 Å map. Duplicate particles were removed using a minimum separation distance of 80 Å, and the remaining particles (108k) were subjected to a non-uniform refinement (*Punjani et al., 2020*) with C2 symmetry imposed and per-particle defocus and CTF parameter optimization enabled, which yielded a 3.57 Å map.

To employ Bayesian polishing in RELION-3 (*Zivanov et al., 2018*), the same 432 micrographs were motion-corrected in RELION 3.1 using its own implementation of MotionCorr2 (*Zheng et al., 2017*) with 5 by 5 patches. The particle.cs file from the cryoSPARC non-uniform refinement job was converted to star format using pyem (*Asarnow et al., 2019*) with the micrograph path modified to that of the RELION motion-corrected micrographs. Particles were then re-extracted from RELION motion-corrected micrographs with a box size of 256 pixels. Due to the failure of RELION 3D auto-refine to converge on a reasonable structure from these particles, particles were imported back into cryoSPARC and subjected to homogeneous refinement. The particle.cs file from this refinement job was again converted to a star file using pyem. Relion_reconstruct was employed to reconstruct two half maps using the offset and angle information refined in cryoSPARC with the same half data split, and post-processing was performed on the resulting half maps using the refinement mask from cryoSPARC. Bayesian polishing (*Zivanov et al., 2019*) was then performed with this post-processing job as input. The resulting shiny particles were successfully refined to a 3.85 Å map with 3D auto-refine in RELION. After one round of CTF refinement (*Zivanov et al., 2020*), the particles were subjected to another round of Bayesian polishing, and the resulting shiny particles were imported back into cryoSPARC. Non-uniform refinement on shiny particles with C2 symmetry imposed and per-particle defocus and CTF parameter optimization enabled yielded the final 3.46 Å resolution map used for model building and analysis.

## AlphaFold prediction and atomic model building

The sequence for the *Synechococcus* phage S-CBP4 α subunit was retrieved from UniProt (***UniProt Consortium, 2021***) with accession number M1PRZ0. The sequence was used as input for AlphaFold2 prediction (***Jumper et al., 2021***) with the five default model parameters and a template date cutoff of May 14, 2020. As the five models were largely identical in the core region and differing only in the location of the C-terminal tail, the structure predicted with the first model parameter was used in the subsequent process.

The predicted structure of the *Synechococcus* phage S-CBP4 α subunit was first processed and docked into the unsharpened cryo-EM map in PHENIX (***Liebschner et al., 2019***). The 25 N-terminal residues and 45 C-terminal residues were then manually removed due to lack of cryo-EM density, and residues 26–426 were retained in the model. We observed unmodeled density at the specificity site, and based on solution composition, we modeled a TTP molecule. A structure of TTP coordinating a magnesium ion was extracted from the crystal structure of *Bacillus subtilis* RNR (pdb: 6mt9) (***Thomas et al., 2019***) and rigid-body fit into the unmodeled density in Coot (***Emsley and Cowtan, 2004***). The combined model was refined with unsharpened and sharpened maps using phenix.real_space_refine (***Afonine et al., 2018***; ***Liebschner et al., 2019***), with a constraint applied on the magnesium ion coordinated by the triphosphate in TTP according to the original configuration. Residue and loop conformations in the resulting structure were manually adjusted in Coot to maximize fit to map and input for an additional round of real-space refinement in PHENIX with an additional restraint for the disulfide bond between C30 and C196. The atomic coordinates and maps have been deposited to the Protein Data Bank and EM Data Bank under accession codes 7urg and EMD-26712. Due to the weak density for the magnesium ion, it was removed from the atomic coordinates when deposited into PDB.

## Acknowledgements

The authors are grateful to Drs. Will Thomas and Steve Meisburger for helpful discussions and Drs. Richard Gillilan and Qingqiu Huang for assistance at the 1D7A beamline at CHESS. SAXS was conducted at the Center for High Energy X-ray Sciences (CHEXS), which is supported by the National Science Foundation (NSF) under award DMR-1829070, and the Macromolecular Diffraction at CHESS (MacCHESS) facility, which is supported by award 1-P30-GM124166-01A1 from the National Institute of General Medical Sciences (NIGMS), National Institutes of Health (NIH), and by New York State's Empire State Development Corporation (NYSTAR). Cryo-EM work was done using the Cornell Center for Materials Research (CCMR) Shared Facilities, which are supported through the NSF MRSEC program (DMR-1719875). This project was undertaken with the assistance of resources and services from the National Computational Infrastructure (NCI), which is supported by the Australian Government, as well as with the BioHPC resource at the Cornell Institute of Biotechnology. We acknowledge the ARC Centre of Excellence for Innovations in Peptide and Protein Science (CE200100012), the ARC Centre of Excellence in Synthetic Biology (CE200100029). This work was supported by NSF grant MCB-1942668 (to NA) and startup funds from Cornell University (to NA).

## Additional information

### Funding

| Funder | Grant reference number | Author |
| --- | --- | --- |
| National Science Foundation CAREER | MCB-1942668 | Nozomi Ando |
| ARC Centre of Excellence in Synthetic Biology | CE200100029 | Colin J Jackson |
| ARC Centre of Excellence for Innovations in Peptide and Protein Science | CE200100012 | Colin J Jackson |

The funders had no role in study design, data collection and interpretation, or the decision to submit the work for publication.

## Author contributions

Andrew A Burnim, Performed bioinformatic analyses, Performed bioinformatic analyses, Performed bioinformatic analyses, Performed bioinformatic analyses, Performed bioinformatic analyses, Performed bioinformatic analyses; Matthew A Spence, Validation, Devised bioinformatic workflow, Performed computational experiments, Performed data analysis, Assisted manuscript preparation, Edited the manuscript; Da Xu, Performed AlphaFold predictions; Colin J Jackson, Acquired funding, Conceived of the project, Designed and supervised research, Analyzed data, Edited the manuscript; Nozomi Ando, Oversaw the project, Acquired funding, Conceived of the project, Designed and supervised research, Analyzed data, Wrote the paper

## Author ORCIDs

Andrew A Burnim ⓘ https://orcid.org/0000-0002-9962-1397
Matthew A Spence ⓘ http://orcid.org/0000-0003-2284-2498
Da Xu ⓘ http://orcid.org/0000-0002-3879-039X
Colin J Jackson ⓘ https://orcid.org/0000-0001-6150-3822
Nozomi Ando ⓘ https://orcid.org/0000-0001-7062-1644

## Decision letter and Author response

Decision letter https://doi.org/10.7554/eLife.79790.sa1
Author response https://doi.org/10.7554/eLife.79790.sa2

---

# Additional files

## Supplementary files

MDAR checklist

## Data availability

The cryo-EM map has been deposited in the Electron Microscopy Data Bank under accession code EMD-26712, and the model has been deposited in the Protein Data Bank under accession code 7urg. The phylogeny shown in Figure 2 is available at (https://itol.embl.de/shared/yFvz6aVgum9z). The structure-guided sequence alignment and all twenty inferred phylogenies are available for download as supplementary materials.

The following datasets were generated:

| Author(s) | Year | Dataset title | Dataset URL | Database and Identifier |
|---|---|---|---|---|
| Xu D, Burnim AA, Ando N | 2022 | cryo-EM structure of ribonucleotide reductase from Synechococcus phage S-CBP4 bound with TTP | https://www.ebi.ac.uk/emdb/EMD-26712 | EM DataBank, EMD-26712 |
| Xu D, Burnim AA, Ando N | 2022 | cryo-EM structure of ribonucleotide reductase from Synechococcus phage S-CBP4 bound with TTP | https://www.rcsb.org/structure/7URG | RCSB Protein Data Bank, 7URG |

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
