## [Editor Report]

The largest phylogeny of ribonucleotide reductases (RNRs) to-date is presented in this study via combining phylogenetic reconstruction, structural determination and analysis, and extensive discussion on the possible evolutionary history. A new clade of RNRs is uncovered and systematically compared with other well-known lineages of RNRs. Summarizing all observations and integrating previous knowledge, the authors propose a reasonable and not over-interpreted evolutionary model to depict the history of this ancient protein family. In sum, the arguments and data offered throughout this manuscript are strong and convincing, and they can help the community to comprehend the evolution of the diverse and crucial RNRs from a deep perspective.

---

## [Decision Letter]

**Decision letter after peer review:**

Thank you for submitting your article "Comprehensive phylogenetic analysis of the ribonucleotide reductase family reveals an ancestral clade and the role of insertions and extensions in diversification" for consideration by *eLife*. Your article has been reviewed by 3 peer reviewers, including Nir Ben-Tal as the Reviewing Editor and Reviewer #1, and the evaluation has been overseen by a Reviewing Editor and Detlef Weigel as the Senior Editor. The following individuals involved in review of your submission have agreed to reveal their identity: JoAnne Stubbe (Reviewer #2); Anders Hofer (Reviewer #3).

Essential revisions:

Presentation:

1. Through their careful analysis, the authors learned much additional information about RNRs, helping the community to think about the evolutionary question of RNR classes in unique ways (pages 1-9.5).

In pages 9.5 to 21, the paper enters into additional fascinating, but complex analyses. The authors focus on the origin of the 100 amino acid ATP cone domain involved in regulation of RNR activity in distinct ways which are found in all three RNR classes (p 9.5-14). The cone domain is also found in NrdR transcription factors. The domain story is very interesting, but its function and nucleotide regulation still has many unresolved issues. Their analysis as presented, changes our thinking about this domain and its role in RNR which controls their activity by altering the proteins´ quaternary structures (organism specific). Their results, however, while interesting, become complex very rapidly and hard to digest to those uninitiated in the background literature. Perhaps this analysis should be in an independent publication with additional background information.

The second and third topics introduced (14-17 and 17-19) involve the C-terminal domains of the class II RNRs and the issue of the finger loop-motif in the class III RNRs, respectively. Both topics are also very interesting from the discoveries the authors have made in their new analysis approach. In these sections as well, more background for those uninitiated in RNR is required. We feel that it would make the manuscript easier to read, if these parts are saved for an additional publication.

In summary, we favor a division of this manuscript into two or more papers. Their approach and results (1-9.5) provide us with a unique and important picture that will be of general interest. One additional suggestion might make it easier for readers uninitiated in the complexity of RNRs. In the introduction, insert an additional figure. Some variation on Figure 1(a,b,c) in a recent review in Ann Rev Biochem (2020, Greene et al) could be very helpful. Also, in the first 9.5 pages make clearer to the reader the issues of simple sequence alignments and evolution (horizonal gene transfer etc), the power of the author's approach and also perhaps a critical evaluation of the approach.

2. The authors analyzed the N-terminal ATP-cone domain of all RNRs, long C-termini of class II RNRs and the finger-loop-motif of class III RNRs separately after getting the phylogeny of RNR. The first weakness of these descriptions is that the authors didn't clearly describe what was known before and what was newly uncovered by this study. It would be better if the authors can stress what is unknown before this report when depicting the evolution of these insertions and extensions. Furthermore, although well-written in general, description on these regions is a bit scattered, making it difficult for readers to grasp the points. Is it possible to add an illustration or a table to summarize all classifications of these three insertions/extensions regions?

3. Abstract (first sentence): There are actually a few organisms that lack RNR and they are all pathogens, including Borrelia burgdorferi, Giardia intestinalis and Ureaplasma urealyticum. A workaround could be to replace "all organisms" with "all free-living organisms" since all of the ones lacking the enzyme live inside a host organisms.

4. Introduction (first paragraph). Same issue as in the abstract. Perhaps it is possible to also mention some of these exceptions here.

5. Move the description of the phylogenetical subclassification (NrdAe, NrdAg etc) to the introduction (now it is not mentioned until p 11). As it is written now, it is confusing that only the class Ia-e classification is mentioned early although the phylogenetic classification is more relevant here where the main focus is on the catalytic subunit. The class Ia-d(e) classification is based on the small subunit mechanism (radical and metal center, see comment above on a metallo-cofactor figure), rather than phylogeny. It would probably be best to mention that there are two parallel subclassification systems early to make everything more clear for the reader.

6. Please add reference for NrdAy.

7. Figure 6B. The class Ib, Ic and Ie subclasses are all included in the tree but not class Id Please add an explanation of that in the figure legend.

Science:

(1) Although it is a very interesting study that opens new perspectives in RNR evolution, we think that it is important to also be honest about that it has both strengths and weaknesses and that we cannot be sure which will be the correct model in the end. As mentioned in the manuscript many of the conclusions are different compared to those in previous studies performed by Britt-Marie Sjöberg´s group regarding positioning of classes and subclasses as well as ATP cone evolution. Generally, all evolutionary studies are based on assumptions, and this is no exception. For example, midpoint rooting is used as a basis to form the evolutionary tree here (Figure 2), and the assumption is then that evolution has a constant speed. Other weaknesses are that the new class is not separate in all presented trees (Figure 2-supplement 1), and that we do not know how the initial assumptions affect further conclusions such as ATP cone evolution, subclassification etc. Nevertheless, we think it is a very interesting study which introduces novel thinking. However, since many conclusions are so different compared to the previous ones, we think it is necessary to emphasize that it is still a model and future studies will gradually give us more and more information about which evolutionary scenario is the correct one. It can rather be seen as an interesting starting point for extended evolutionary studies by the RNR research community.

(2) The authors arbitrarily select a sequence identity threshold 85% in the initial sequence collection for sequence similarity network analysis and subsequent phylogenetic reconstruction. This specific choice needs to be justified. How robust is the workflow to thresholds?

(3) In the phylogenetic inference and evo-velocity analysis, the authors found that class I and class II RNRs are joint over the same sequence space. The ATP-cone sequences of class II RNR, nevertheless, were clustered with that of class III RNR, suggesting a common origin of ATP-cone of these two clades. The authors propose a scenario where most of class II RNRs lost their ATP-cone domains during evolution. In this scenario, class II RNRs are evolutionarily closer to class I RNRs after the divergence, but their ATP-cone sequences keep a higher similarity with that of class III RNRs, instead of class I RNRs. Could this interesting finding be explained more? Have the authors attempted to perform ancestral sequence reconstruction on this region? Is it possible that the ATP-cone sequences of class II/III RNRs are more ancient? Namely, that there is lower evolutionary pressure to drive the evolution of ATP-cone domains in class II and class III RNRs, and thus make their homology more significant.

(4) The authors reported that a structural motif observed in class II C-termini are structurally homologous with the iron-sulfur cluster assembly protein IscU. Is this evolutionary link known before? Does this homology represent a potential evolutionary clue between two protein families, like the recent bridging theme definition proposed by Kolodny et al. (PMID: 33502503)?

(5) Text associated with Figure 2. Describe the basic assumption with midpoint rooting with strength and weaknesses.

(6) Figure 4 (supp 5). The C-terminus of the R2 subunit looks very short, is it long enough to bind the R1 subunit?

(7) Page 3. It is mentioned that all RNRs have specificity regulation. However, RNRs from the Herpesviridae family lack this regulation.

(8) Page 10. The last common ancestor didn't necessarily have had an ATP cone. Many species have more than one class of RNR and since it is a mobile element it can easily jump from one class to another. It can also be argued that most of class II, class φ, the most deeply rooted class I (Ai, Ak and E) and many class III species lack the ATP cone. It would be good to give a balanced view of pros and cons to this theory.

(9) Page 21. How did limited phosphorous supplies have led to the loss of the ATP cone? First, the logic behind it is unclear: do you mean that less phosphate gives less ribonucleotide substrates to use? Second, losses of the ATP cone have occurred multiple time in the evolution regardless of phosphorous supplies.

(10) Figure 6. It is surprising that there are no sequences with multiple ATP cones except in NrdAz in the tree. There should be several in NrdAh, a few in NrdAe and some NrdD as well.

(11) Figure 6-supplement 1. Do you have any information about where NrdAy ends up in the right tree?

Many of the above comments speak to the complexity of RNRs because of insufficient background summary on RNRs and the challenges with integration of the first and last parts of the paper.

---

## [Author Response]

Essential revisions:Presentation:1. Through their careful analysis, the authors learned much additional information about RNRs, helping the community to think about the evolutionary question of RNR classes in unique ways (pages 1-9.5).In pages 9.5 to 21, the paper enters into additional fascinating, but complex analyses. The authors focus on the origin of the 100 amino acid ATP cone domain involved in regulation of RNR activity in distinct ways which are found in all three RNR classes (p 9.5-14). The cone domain is also found in NrdR transcription factors. The domain story is very interesting, but its function and nucleotide regulation still has many unresolved issues. Their analysis as presented, changes our thinking about this domain and its role in RNR which controls their activity by altering the proteins´ quaternary structures (organism specific). Their results, however, while interesting, become complex very rapidly and hard to digest to those uninitiated in the background literature. Perhaps this analysis should be in an independent publication with additional background information.The second and third topics introduced (14-17 and 17-19) involve the C-terminal domains of the class II RNRs and the issue of the finger loop-motif in the class III RNRs, respectively. Both topics are also very interesting from the discoveries the authors have made in their new analysis approach. In these sections as well, more background for those uninitiated in RNR is required. We feel that it would make the manuscript easier to read, if these parts are saved for an additional publication.In summary, we favor a division of this manuscript into two or more papers. Their approach and results (1-9.5) provide us with a unique and important picture that will be of general interest. One additional suggestion might make it easier for readers uninitiated in the complexity of RNRs. In the introduction, insert an additional figure. Some variation on Figure 1(a,b,c) in a recent review in Ann Rev Biochem (2020, Greene et al) could be very helpful. Also, in the first 9.5 pages make clearer to the reader the issues of simple sequence alignments and evolution (horizonal gene transfer etc), the power of the author's approach and also perhaps a critical evaluation of the approach.

We thank the reviewers for the positive and comprehensive feedback. As suggested, we have divided the manuscript into two papers (see color-coded version of the originally submitted manuscript), and our revised manuscript is now focused on p. 1-9.5 of our original submission. As suggested, we have added a new introductory figure (Figure 1 —figure supplement 1) that summarizes the RNR mechanism and different classes. We have added a critical evaluation of evolutionary models both in the Introduction (highlighted in yellow on p. 3-4) and the Discussion (highlighted in yellow on p. 10-11).

In addition to addressing all reviewer comments, the revision also includes an expanded discussion of class Ø RNRs. RNRs near the active site (Figure 4 —figure supplement 5). We have also collected new SAXS data such that the MgCl2 concentration is fixed, rather than changing with nucleotide concentration (Figure 4D and Figure 4 —figure supplement 2); conclusions remain the same.

2. The authors analyzed the N-terminal ATP-cone domain of all RNRs, long C-termini of class II RNRs and the finger-loop-motif of class III RNRs separately after getting the phylogeny of RNR. The first weakness of these descriptions is that the authors didn't clearly describe what was known before and what was newly uncovered by this study. It would be better if the authors can stress what is unknown before this report when depicting the evolution of these insertions and extensions. Furthermore, although well-written in general, description on these regions is a bit scattered, making it difficult for readers to grasp the points. Is it possible to add an illustration or a table to summarize all classifications of these three insertions/extensions regions?

We appreciate this comment. As suggested, we have split off p. 9.5-14 into a separate manuscript, in which will be able to provide more in-depth background on these topics. However, we have also added a table to the current revised manuscript as Figure 1 —figure supplement 1.

3. Abstract (first sentence): There are actually a few organisms that lack RNR and they are all pathogens, including Borrelia burgdorferi, Giardia intestinalis and Ureaplasma urealyticum. A workaround could be to replace "all organisms" with "all free-living organisms" since all of the ones lacking the enzyme live inside a host organisms.

We agree with the reviewer and have changed the wording as suggested on p. 2.

4. Introduction (first paragraph). Same issue as in the abstract. Perhaps it is possible to also mention some of these exceptions here.

We agree with the reviewer and have changed the wording as suggested on p. 1.

5. Move the description of the phylogenetical subclassification (NrdAe, NrdAg etc) to the introduction (now it is not mentioned until p 11). As it is written now, it is confusing that only the class Ia-e classification is mentioned early although the phylogenetic classification is more relevant here where the main focus is on the catalytic subunit. The class Ia-d(e) classification is based on the small subunit mechanism (radical and metal center, see comment above on a metallo-cofactor figure), rather than phylogeny. It would probably be best to mention that there are two parallel subclassification systems early to make everything more clear for the reader.

We thank the reviewer for this feedback. The Nrd-based classification system is not used in the revised manuscript, but we plan to introduce the two parallel subclassification systems in the introduction of the second paper (covering the topics in p. 9.5-14 of the original submission).

6. Please add reference for NrdAy.

This phylogenetic subclass was introduced in the RNRdb (Lundin, et al. 2009). NrdAy sequences from the RNRdb were included in the full dataset of Martínez-Carranza et al. (2020), but were filtered in the final, pared down dataset that was described in the paper. We will cite these works in the second paper.

7. Figure 6B. The class Ib, Ic and Ie subclasses are all included in the tree but not class Id Please add an explanation of that in the figure legend.

We thank the reviewer for this feedback. Although this figure is not part of the revised manuscript, we will revise the figure and legend for the second paper (covering the topics in p. 9.5-14).

Science:(1) Although it is a very interesting study that opens new perspectives in RNR evolution, we think that it is important to also be honest about that it has both strengths and weaknesses and that we cannot be sure which will be the correct model in the end. As mentioned in the manuscript many of the conclusions are different compared to those in previous studies performed by Britt-Marie Sjöberg´s group regarding positioning of classes and subclasses as well as ATP cone evolution. Generally, all evolutionary studies are based on assumptions, and this is no exception. For example, midpoint rooting is used as a basis to form the evolutionary tree here (Figure 2), and the assumption is then that evolution has a constant speed. Other weaknesses are that the new class is not separate in all presented trees (Figure 2-supplement 1), and that we do not know how the initial assumptions affect further conclusions such as ATP cone evolution, subclassification etc. Nevertheless, we think it is a very interesting study which introduces novel thinking. However, since many conclusions are so different compared to the previous ones, we think it is necessary to emphasize that it is still a model and future studies will gradually give us more and more information about which evolutionary scenario is the correct one. It can rather be seen as an interesting starting point for extended evolutionary studies by the RNR research community.

We agree with the reviewers that all evolutionary models are models that are based on assumptions. We have updated the text on p. 3-4 (yellow highlights) to clarify that large-scale phylogenetic inference was not possible prior to our work and have included a critical discussion of phylogenetic methods and evolutionary models in the Discussion on p. 10-11 (yellow highlights). We address each point raised below:

We thank the reviewer for the opportunity to discuss the observation of the 2/20 inferred trees that do not show the class Ø as ancestral to the class I and II clade. In these two trees, the class Ø clade instead diverges out of the class II major clade. Each of the 20 inferred phylogenies fails to be rejected by the approximately unbiased test and thus are all valid hypotheses. However, when comparing the log likelihoods in topology testing of each tree, we find that these two trees (tree 6 and 7) are the least likely of those inferred with the LG+R10 model. We have updated the text on p. 5 to reflect this observation.To be as rigorous as possible, we performed 20 replicates of tree inference using two different sets of assumptions (models of evolution). For a sequence set of nearly 7000 sequences, this required 1.4 million CPU hours and 7 continuous months of wall time. However, as suggested below, we also performed phylogenetic inference with the sequence redundancy changed from 85% to 55%. The topology with the highest statistical support (i.e., that with class Ø as ancestral to the class I and II clade) was reproduced 100% independently of the trees which we originally presented. This new result is described on p. 5-6 and Figure 2 —figure supplement 2.We have revised the text on p. 6 to clarify our choice of mid-point rooting. We further show that the locations of the midpoint in the full-dataset phylogeny and the reduced redundancy phylogeny (Figure 2 —figure supplements 1 and 2) are consistent, further validating the choice to root on the midpoint.

(2) The authors arbitrarily select a sequence identity threshold 85% in the initial sequence collection for sequence similarity network analysis and subsequent phylogenetic reconstruction. This specific choice needs to be justified. How robust is the workflow to thresholds?

Phylogenetic reconstruction was repeated with the sequence set to 55%. The topology with the highest statistical support (i.e., that with class Ø as ancestral to the class I and II clade) is reproduced 100% independently of the trees which we originally presented. We believe this result clearly demonstrates how robust our conclusions are to the redundancy threshold and taxon sampling. We have described this in the Results on p. 5-6 and in Figure 2 —figure supplement 2 and have updated the Methods on p. 13-14.

(3) In the phylogenetic inference and evo-velocity analysis, the authors found that class I and class II RNRs are joint over the same sequence space. The ATP-cone sequences of class II RNR, nevertheless, were clustered with that of class III RNR, suggesting a common origin of ATP-cone of these two clades. The authors propose a scenario where most of class II RNRs lost their ATP-cone domains during evolution. In this scenario, class II RNRs are evolutionarily closer to class I RNRs after the divergence, but their ATP-cone sequences keep a higher similarity with that of class III RNRs, instead of class I RNRs. Could this interesting finding be explained more? Have the authors attempted to perform ancestral sequence reconstruction on this region? Is it possible that the ATP-cone sequences of class II/III RNRs are more ancient? Namely, that there is lower evolutionary pressure to drive the evolution of ATP-cone domains in class II and class III RNRs, and thus make their homology more significant.

We thank the reviewer for this detailed question, which will be addressed in the second paper (on p. 9.5-14). Briefly, key points are addressed below:

First, we will clarify that we have not done ancestral sequence reconstruction on the N-terminus. To do so would require a phylogeny of the ATP-cones, but the phylogenetic reconstructions that we present are based on the alignment of the core catalytic barrel and do not include the N-termini, which are highly variable in length and therefore do not align. Thus, ancestral sequences for the N-terminal regions cannot be calculated.In response to two other, related questions: Yes, we propose that ATP-cones of class II/III RNRs are more ancient based on their clustering in our SSN, and we hypothesize that there was lower evolutionary pressure to diversify the ATP-cone domains in class II and III RNRs, compared to those of class I RNR (which appear all over the SSN). For the class II RNRs, we hypothesize that the lack of need contributed to the lack of diversification of the ATP-cones. Based on our proposal that there was a common ancestor for all RNR classes with an ATP-cone, it would not be strange for the N-terminal regions of class II/III RNRs to retain homology while the core catalytic barrel diversified.

(4) The authors reported that a structural motif observed in class II C-termini are structurally homologous with the iron-sulfur cluster assembly protein IscU. Is this evolutionary link known before? Does this homology represent a potential evolutionary clue between two protein families, like the recent bridging theme definition proposed by Kolodny et al. (PMID: 33502503)?

We thank the reviewer for this detailed question and reference, which will be addressed in the second paper (on p. 9.5-14). Firstly, this evolutionary link was not observed before for any of the RNRs. Secondly, the bridging themes discussed in Kolodny et al. are defined as short “themes” (similar sequence segments shared between proteins) that are found in different sequential and structural contexts. This concept applies to short sequences based on sequence similarity rather than structural similarity. The IscU domain is significantly longer (~130 aa) than the bridging themes discussed in the work of Kolodny et al. (20-80 aa). In addition, an all-vs-all pBLAST analysis between class II C-terminal tail and IscU family (PF01592) shows no hits with an E-value lower than 1E-5, indicating poor sequence similarity between the two, though AlphaFold predicts a similar fold. Also, due to the exclusiveness of this IscU-like domain in class II C-terminal region, it is unlikely that the RNR catalytic core has evolutionary relatedness to the IscU family. As stated in the main text, we believe a likely evolutionary scenario for the acquisition of the IsuU-like domain is through the insertion of an IscU-like protein into the class II RNR C-terminus at some point during its evolutionary history.

(5) Text associated with Figure 2. Describe the basic assumption with midpoint rooting with strength and weaknesses.

As described in response to (1) and (2), we have revised the text on p. 6.

(6) Figure 4 (supp 5). The C-terminus of the R2 subunit looks very short, is it long enough to bind the R1 subunit?

The text has been revised on p. 10 to clarify that the ferritin-like protein cannot bind the α subunit in the way that class I subunits interact (via the C-terminus of the R2 subunit) and that this does not preclude the possibility that two class Ø subunits interact in a different manner.

(7) Page 3. It is mentioned that all RNRs have specificity regulation. However, RNRs from the Herpesviridae family lack this regulation.

We thank the reviewer for this detail. The Introduction has been revised on p. 2-3 (highlighted in cyan) as the original text was intended to provide background for the content on p. 9.5-14.

(8) Page 10. The last common ancestor didn't necessarily have had an ATP cone. Many species have more than one class of RNR and since it is a mobile element it can easily jump from one class to another. It can also be argued that most of class II, class φ, the most deeply rooted class I (Ai, Ak and E) and many class III species lack the ATP cone. It would be good to give a balanced view of pros and cons to this theory.

We thank the reviewer for this question, which will be addressed in the second paper (on p. 9.5-14). We would first like to clarify that we are not aware of any direct evidence that the ATP-cone is a mobile element in the sense that it can jump from one class to another. The ATP-cone was first called an “evolutionarily mobile element” in the work of Aravin and Koonin. This term, defined in a previous work (Doolittle & Bork, Sci Am 1998), refers to motifs that have been found in multiple protein families and does not imply that they jump within a protein family. It has been shown by multiple groups that the ATP-cone motif is indeed prevalent in the RNR family, but we are not aware of evidence that this was the result of horizontal gene transfer.

Our work is the first to examine the subclasses of extant ATP-cones, and based on our results, we argue that the ATP-cone is less likely to be a genetically mobile element than it is an evolutionarily fixed domain. Although many organisms do have multiple classes of RNRs within them, the different classes of RNR do not necessarily share the same evolutionary pressure for allosteric regulation, and hence can benefit from the loss of the ATP-cone domain. However, we agree that our discussion can be aided with a more balanced view of ATP-cone evolution due to the many possible scenarios that have resulted in the current sequence landscape of ATP-cones. We plan to expand on our findings in greater detail in a future publication.

(9) Page 21. How did limited phosphorous supplies have led to the loss of the ATP cone? First, the logic behind it is unclear: do you mean that less phosphate gives less ribonucleotide substrates to use? Second, losses of the ATP cone have occurred multiple time in the evolution regardless of phosphorous supplies.

We thank the reviewer for this question, which will be addressed in the second paper (on p. 9.5-14). This comment references the hypothesis of how nutrient starved organisms, such as cyanophages, contain minimal genomes as discussed in Gao et al. 2016 as an explanation for the size of the class Ø sequences. Our hypothesis was that the ATP-cone would be lost, both to minimize the size of the genome and because there would likely be no pressure to maintain allosteric regulation in a nutrient deprived ecological niche.

(10) Figure 6. It is surprising that there are no sequences with multiple ATP cones except in NrdAz in the tree. There should be several in NrdAh, a few in NrdAe and some NrdD as well.

The reviewer is correct, and the figure includes these but are occluded by other symbols. The figure has been revised for the second paper to make sure these are visible.

(11) Figure 6-supplement 1. Do you have any information about where NrdAy ends up in the right tree?

NrdAy is a very small subclade that was first described in the RNRdb. It was included in the full dataset used in the work of Martínez-Carranza et al., but it was filtered from the final dataset that was presented the paper. Our work is the first to show this subclade in a phylogenetic tree.

Many of the above comments speak to the complexity of RNRs because of insufficient background summary on RNRs and the challenges with integration of the first and last parts of the paper.

We thank the reviewers for the above comments. We have addressed these issues by focusing this manuscript completely on the observation of class Ø in our phylogenetic inference and by the addition of new figures and expanded discussions.